# Bistable organic electrochemical transistors: enthalpy vs. entropy

Lukas M. Bongartz [1] ✉, Richard Kantelberg [1], Tommy Meier[1], Raik Hoffmann[2], Christian Matthus[3], Anton Weissbach [1], Matteo Cucchi[1], Hans Kleemann [1] & Karl Leo[1]

Organic electrochemical transistors (OECTs) underpin a range of emerging technologies, from bioelectronics to neuromorphic computing, owing to their unique coupling of electronic and ionic charge carriers. In this context, various OECT systems exhibit significant hysteresis in their transfer curve, which is frequently leveraged to achieve non-volatility. Meanwhile, a general understanding of its physical origin is missing. Here, we introduce a thermodynamic framework that readily explains the emergence of bistable OECT operation via the interplay of enthalpy and entropy. We validate this model through temperature-resolved characterizations, material manipulation, and thermal imaging. Further, we reveal deviations from Boltzmann statistics for the sub-threshold swing and reinterpret existing literature. Capitalizing on these findings, we finally demonstrate a single-OECT Schmitt trigger, thus compacting a multi-component circuit into a single device. These insights provide a fundamental advance for OECT physics and its application in non-conventional computing, where symmetry-breaking phenomena are pivotal to unlock new paradigms of information processing.

Organic electrochemical transistors (OECTs) are at the foundation of numerous emerging technologies, ranging from bioelectronic implants[1,2] to analog neuromorphic computing[3–6], and have recently also expanded into the realm of complementary circuitry[7]. This track record is tied to a unique switching mechanism, based on the coupling of ionic and electronic charge carriers in an organic mixed ionic-electronic conductor (OMIEC)[8–10]. Connected with two electrodes, source and drain, polarons can be driven along the channel by applying a corresponding voltage $V_{DS}$, giving rise to a drain current $I_D$. By immersing the system in an electrolyte along with a gate electrode, a second voltage $V_{GS}$ enables control over the ion flow between electrolyte and channel, and as such, the doping level of the latter (Fig. 1d). The benchmark channel material of OECTs is the polymer blend poly(3,4-ethylenedioxythiophene) polystyrene sulfonate (PEDOT:PSS), where the transistor switching

can be described by

$$PEDOT:PSS + C^+ + A^- \rightleftharpoons PEDOT^+:PSS + C^+ + A^- + e^-, \quad (1)$$

with PEDOT:PSS and PEDOT⁺: PSS as the initial and doped state of the OMIEC. $C^+$ and $A^-$ are the electrolyte cat- and anions and most commonly are given by liquid electrolytes such as aqueous NaCl solution. For a long time, OECTs have been understood by concepts borrowed from field-effect transistor (FET) theories, in particular based on the foundational work by Bernards and Malliaras[11]. At the same time, however, charge formation and transport in FETs differ fundamentally from those in OECTs. While in the former, charges are induced at the semiconductor interface and form a quasi-2D layer, doping in OECTs happens throughout the entire channel, where the formation of electrical double layers at the polymer strands results from an

[1]IAPP Dresden, Institute for Applied Physics, Technische Universität Dresden, Nöthnitzer Str. 61, 01187 Dresden, Germany. [2]Fraunhofer Institute for Photonic Microsystems IPMS, Center Nanoelectronic Technologies, An der Bartlake 5, 01099 Dresden, Germany. [3]Chair of Circuit Design and Network Theory (CCN), Faculty of Electrical and Computer Engineering, Technische Universität Dresden, Helmholtzstr. 18, 01069 Dresden, Germany. ✉e-mail: lukas.bongartz@tu-dresden.de

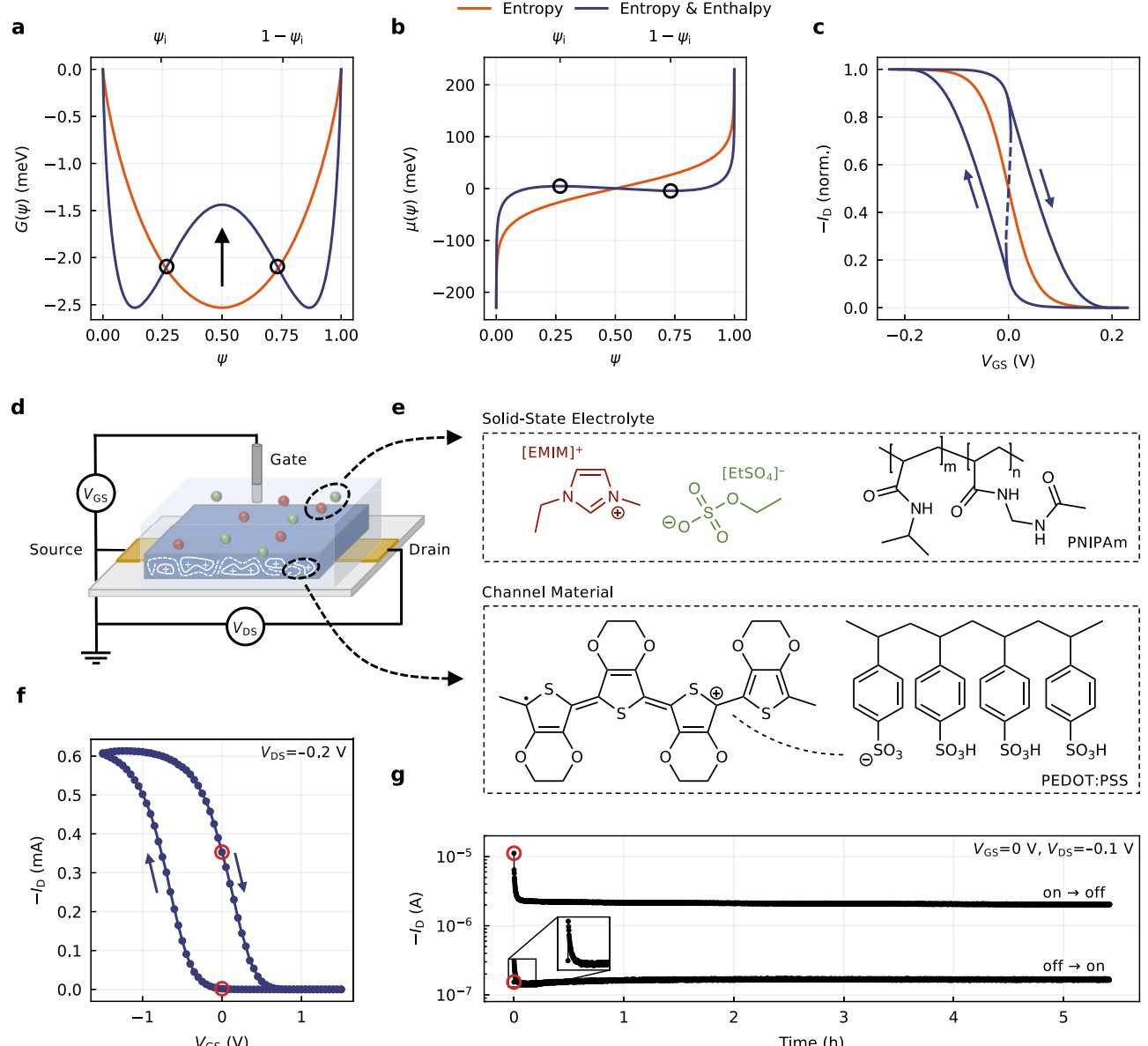

**Fig. 1 | Bistability emerges from the interplay of enthalpy and entropy. a–c** The Gibbs free energy function $G(\psi)$ governs the chemical potential $\mu(\psi)$, which itself determines the transfer curve of an OECT. In the case of dominating entropy, the system is monostable along the $\psi$-axis, with a monotonic chemical potential and an accordingly shaped transfer curve (orange). For rising enthalpy, the system gets bistable, causing a non-monotonic chemical potential, which results in a bistable switching behavior (blue). **d** Generalized setup of an OECT. Devices in this work typically employed a side-gate architecture (Fig. S2). **e** Chemical structures of the OECT solid-state electrolyte and channel material used to study the bistability, with **f** a representative, experimental transfer curve ($V_{DS} = -0.2$ V). **g** Two distinct drain current levels are found when operating the OECT and holding a 0 V gate bias, approaching either from the on- or the off-state.

electrochemical process[12]. Cucchi et al. have set out to resolve this mismatch by describing OECTs in terms of thermodynamics, thereby addressing the long-standing question of the non-linear voltage dependence of the capacitance[13]. In particular, this work describes the process of Eq. (1) as being driven by entropy, where the gate voltage $V_{GS}$ determines the electrochemical potential and as such shifts the equilibrium charge carrier concentration in PEDOT. However, these considerations are limited to the unidirectional operation of the OECT and pertain to the edge case where minimal or no interactions are assumed.

In this work, we move beyond this limitation. We consider the interactions involved in the doping cycle in terms of enthalpy, which allows us to reveal the peculiar situation of a bistable switching behavior. We do so by moving away from aqueous electrolytes to a solid-state system, in order to expose otherwise screened interactions and their effect on the Gibbs free energy. We establish a mathematical condition for the degree of bistability and underpin our reasoning with three sets of experimental evidence. Among others, we show the exceptional scenario in which the subthreshold swing behaves contrary to what would be expected from Boltzmann's law. We reinterpret results from the literature in the context of this framework, which further supports our model and allows for an alternative view of material properties for neuromorphic applications. Finally, our insights culminate in demonstrating how the bistable switching behavior can be leveraged to realize the functionality of a Schmitt trigger—a multi-component circuit—through a single device, laying the foundation for more complex circuitry in neuromorphic and asynchronous computing.

## Results
### Theoretical framework
The Gibbs free energy is defined as the difference between enthalpy and entropy scaled by temperature, according to

$$G = H - TS, \tag{2}$$

where $H$ is enthalpy, $T$ is temperature, and $S$ is entropy. Akin to Cucchi et al.[13], we consider the Gibbs free energy as a function of an intensive state variable, here being the doping parameter $\psi$. Let the channel be composed of doping subunits, where one subunit is the smallest entity that satisfies Eq. (1), then $\psi$ describes the ratio of doped units ($N_{\text{doped}}$) to the total number available ($N_{\text{tot}}$). In other words, it serves as a statistical measure and reflects the probability distribution of doped units across all microstates:

$$\psi = \frac{N_{\text{doped}}}{N_{\text{doped}} + N_{\text{undoped}}} = \frac{N_{\text{doped}}}{N_{\text{tot}}}. \tag{3}$$

The OECT switching can be understood as controlling this very ratio. For such a process, the Gibbs free energy (per unit) follows as

$$G(\psi) = H^0(\psi) + H_{\text{tr}}(\psi) - TS_{\text{tr}}(\psi), \tag{4}$$

where $H^0(\psi)$ is the standard enthalpy. $H_{\text{tr}}(\psi)$ and $S_{\text{tr}}(\psi)$ are the enthalpy and entropy associated with the state transitions of Eq. (1), defined as

$$H_{\text{tr}}(\psi) = \frac{1}{2}\left( h_{dd}\psi^2 + h_{uu}(1-\psi)^2 + 2h_{du}\psi(1-\psi) \right) \quad \text{and} \tag{5}$$

$$S_{\text{tr}}(\psi) = -k_{\text{B}}\left( \psi \ln(\psi) + (1-\psi)\ln(1-\psi) \right). \tag{6}$$

Here, $h_{dd}$, $h_{uu}$, and $h_{du}$ denote the intra- and interspecies interaction strength of doped and undoped sites, and $k_{\text{B}}$ is the Boltzmann constant (see Supplementary Note 1 for a comprehensive derivation). Both, the doping parameter $\psi$ and the Gibbs free energy $G(\psi)$ can be translated into device-level quantities. $\psi$ is proportional to the polaron density and, therefore, electrical conductivity (drain current), while the chemical potential $\mu(\psi)$ links $G(\psi)$ to the gate-source voltage over the electrochemical potential $\bar{\mu}$:

$$\left( \frac{\partial G(\psi)}{\partial \psi} \right)_{p,T} = \mu(\psi) \leftrightarrow \bar{\mu} = \mu(\psi) + fe(V_{\text{GS}} - V_{\text{Ch}}), \tag{7}$$

with $e$ as the elementary charge and $V_{\text{Ch}}$ as the channel potential. $f$ is a fudge factor related to the effective modulation of the chemical potential through the gate-source voltage (doping efficiency) and underlies the translation from the theoretical energy scales (Fig. 1b) to the experimentally found voltages (Fig. 1f). In consequence of Eq. (7), one can understand the switching characteristics of OECTs as a natural consequence of their underlying Gibbs free energy profile.

For a system of negligible interaction (enthalpy), entropy is the underlying driving force. As depicted in Fig. 1a (orange), such systems possess a single minimum in $G(\psi)$, i.e., a single equilibrium state at $\psi_0$. It follows a monotonic chemical potential (Fig. 1b), which translates to the transfer curve and determines the characteristic saturation in the on- and off-states of OECTs (Fig. 1c)[13]. This situation changes distinctively when enthalpic contributions are taken into account. As derived in Supplementary Note 1, the single equilibrium state in $G(\psi)$ bifurcates for

$$\lambda = \frac{(h_{dd} + h_{uu} - 2h_{du})}{k_{\text{B}}T} \cdot \psi(\psi-1) \geq 1 \quad \text{with} \quad \psi \in [\psi_i, 1-\psi_i], \tag{8}$$

where $G(\psi)$ has a negative curvature. Doping concentrations in this $\psi$-range are unstable and break into two coexisting equilibrium states along the doping axis, where $\psi_i$ and $1-\psi_i$ define the local extrema of the non-monotonic chemical potential. This thermodynamic instability goes along with a dynamic instability (Supplementary Note 2), which inevitably suggests a bistable switching behavior (Fig. 1a–c, blue). The quantity $\lambda$ sets the enthalpic and entropic contributions into relation and thus serves as a bifurcation parameter, reflecting the degree of bistability present in the system (Fig. S1). We provide an interactive simulation tool under ref. 14 to illustrate these relationships.

### Theoretical implications
The bistability appears as the result of dominating enthalpy over entropy, which itself originates from the interactions underlying Eq. (5). It is reasonable to assume that their impact is most observable in a non-aqueous system, where dielectric shielding is minimized. At the same time, the ionic species themselves would benefit from a particularly strong dipole moment in order to penetrate and dope the OMIEC in the absence of water. For this purpose, we turn to the previously reported OECT system (Figs. 1e, S2) of PEDOT:PSS and 1-ethyl-3-methylimidazolium ethyl sulfate ([EMIM][EtSO$_4$]) in a matrix of poly(N-isopropylacrylamide) (PNIPAm)[15], offering multiple advantages: First, the solid-state system can be channeled into an inert gas atmosphere without degradation, providing a controlled, water-free environment. Second, [EMIM][EtSO$_4$] offers among the highest dielectric permittivity of commercially available ionic liquids[16], and third, it is known to provide an exceptionally low-lying off-state in OECTs[15], allowing for a particularly large modulation of the charge carrier concentration. That is, a large range over which the $\psi$-axis can be probed.

As demonstrated in previous reports[15,17], this system does, in fact, show a significant hysteresis in its transfer curve (Fig. 1f), which has been shown to remain even for scan rates below $10^{-4}$ V s$^{-1}$ (ref. 15) and is similarly present with a non-capacitive Ag/AgCl-gate electrode (Figs. S3, S4). Both of these findings suggest a reason beyond ion kinetics or capacitive effects[18–20] and motivate to study the presence of coexisting equilibrium states. To this end, we operate an OECT multiple times before holding a 0 V gate bias, approaching either from the on- or the off-state (Fig. S5). As Fig. 1g shows, after a short-stabilizing period (-ms), two distinct drain current levels are present over the course of hours, differing more than one order of magnitude. Crucially, the gate current stabilizes at the same level (-nA) for both conductivity states (Fig. S6). This result clearly indicates the presence of two separate doping states of the channel, present at the same gate bias but originating from different initial states.

We validate the hypothesis of a bistable Gibbs free energy function based on the theoretical consequences that this situation would entail for the ensemble. To this end, three hypotheses are derived (Fig. 2).

To begin with, given that the bistability is attributed to the balance between entropy and enthalpy (Eq. (8)), we conclude that tweaking these quantities should allow for targeted manipulation of the hysteresis in the transfer curve. Both an increasing entropy ($TS_{\text{tr}}$) and a decreasing enthalpy ($H_{\text{tr}}$) should cause the hysteresis to abate, as $\lambda$ decreases (Fig. 2a).

Second, as shown in Supplementary Note 3, it is possible to think of the course of $\mu(\psi)$ as an idealized cycle process built up from processes of $dT = 0$ and $d\mu = 0$. With a non-monotonic chemical potential $\mu(\psi)$, and using the concept of Maxwell constructions, the enclosed area (per switching operation) then corresponds to the chemical work performed ($|W_{\text{Ch}}|$), which in turn should be reflected in an in- and outflow of heat $Q$ (Fig. 2b).

Finally, these considerations can be extended to the subthreshold behavior. With the total chemical potential remaining constant during the transition between the on- and off-state, it follows that

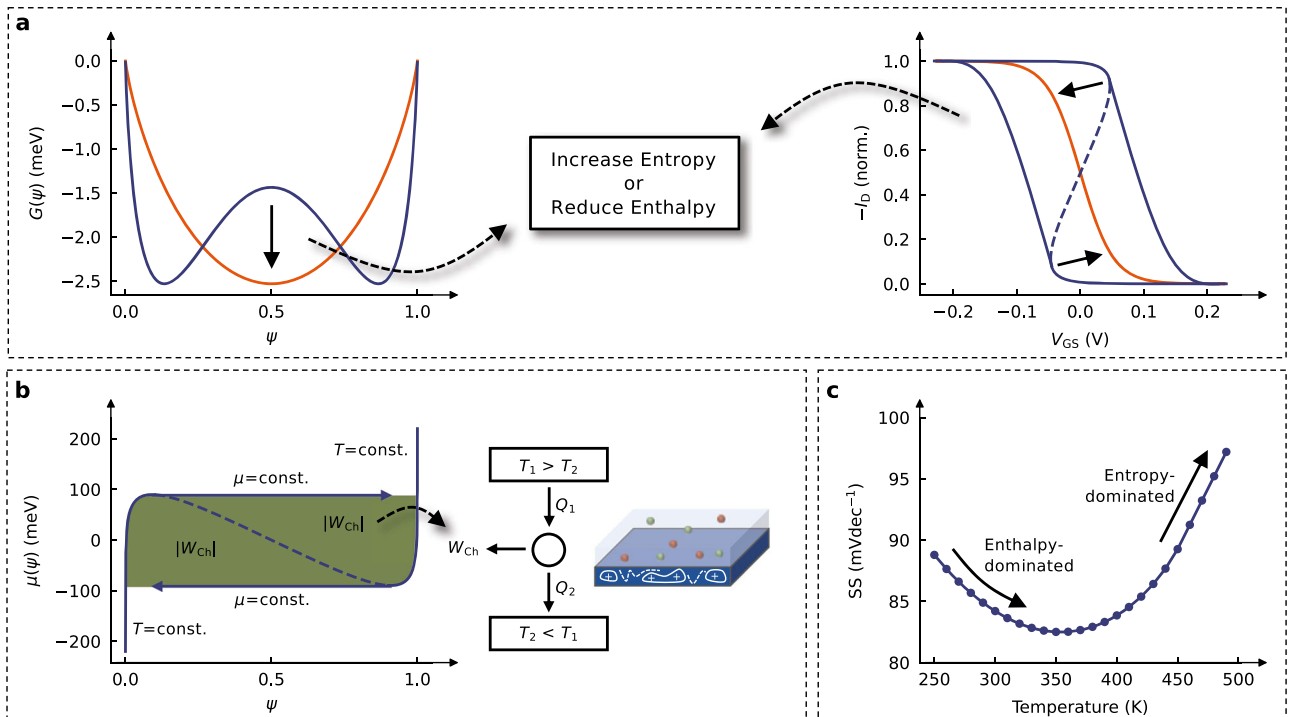

**Fig. 2 | Theoretical implications of a bistable Gibbs free energy function. a** The bistability is expected to decrease for rising entropy ($TS_{tr}$) and falling enthalpy ($H_{tr}$), which should be reflected in the transfer curve by an abating hysteresis. **b** The course of $\mu(\psi)$ can be considered an idealized cycle process, assembled of processes with $dT = 0$ and $d\mu = 0$. Following a Maxwell construction, the integrals correspond to the chemical work performed ($|W_{Ch}|$), from which an in- and outflow of heat $Q$ is to be expected. **c** The bistability suggests a non-monotonic progression of the subthreshold swing (SS) with temperature, where the conventionally expected positive slope only occurs once entropy dominates over enthalpy and the chemical potential is sufficiently monotonic (Eq. (9)).

in this region ($\psi \in [\psi_i, 1-\psi_i]$), the non-monotonic chemical potential counterweights the electrostatic potential from the gate voltage (Eq. (7)). As the non-monotonicity decreases with temperature, this balancing effect is expected to decrease similarly until it vanishes completely once entropy balances out. As shown in Supplementary Note 4, we infer that for a bistable system, such behavior should reflect in the subthreshold swing (SS) being subject to two effects: the counterbalance of the non-monotonic chemical potential as well as the classical influence of diffusion through thermal energy. Since the former is expected to decrease with rising temperature, while the latter scales with $k_B T$, a non-monotonic progression of the subthreshold swing with temperature follows. This relationship can be expressed through

$$\mathrm{SS}(T) \approx \frac{\ln(10)}{e}\left(\left(\left.\frac{\partial G(\psi, T)}{\partial \psi}\right|_{\psi=\psi_i}\right)_{p,T} + k_B T\right), \qquad (9)$$

where the chemical potential is examined at the lower inflection point of $G(\psi)$, i.e., at its local maximum, based on the depletion mode operation of PEDOT:PSS. This equation is modeled in Fig. 2c, showing that the conventionally expected increase of the subthreshold swing with temperature only occurs once the bistability is sufficiently suppressed by entropy. By the same token, we suggest a decreased subthreshold swing for a system of reduced enthalpy, which similarly lessens the non-monotonic nature of the chemical potential profile.

## Experimental validation

We verify these three hypotheses experimentally. To start with, we study the impact of entropy via temperature-dependent transfer measurements in quasi-steady state[21]. As shown in Fig. 3a, these clearly confirm a decreasing bistability for rising temperature, which is also particularly apparent in logarithmic scale (Fig. S7a). To validate the same for lowered enthalpy, we modify the electrolyte system by including KCl as a hygroscopic additive[22], thus shielding the interactions underlying $H_{tr}$. As evidenced in Fig. 3b, this attempt likewise confirms our hypothesis. Strikingly, the modification does not alter the on- or off-state of the OECT (Fig. S7b) and solely causes the hysteresis curve to close. Using this system, we replicate the experiment of Fig. 1g, confirming that the lapsed hysteresis translates to diminished state retention in the long term (Figs. S8 and S9). Note that while the increase in temperature causes a mostly uniform shift in the hysteresis branches as bistability ceases, the experiment involving altered enthalpy primarily affects the dedoping branch. Accordingly, we conclude that the material modification predominantly alters the energetics involved in dedoping, while temperature exerts a more uniform influence on both doping and dedoping.

Normalizing these data sets with respect to $\psi$ and symmetry around $\mu(\psi) = 0$ eV (Fig. 3c), we reconstruct the experimental Gibbs free energy profiles by integration and extract the interaction parameters and doping efficiencies with the methods laid out in Supplementary Notes 5 and 6. As Fig. 3d and e show, both experiments can be traced back to double-minima Gibbs free energy profiles, where either increasing entropy or decreasing enthalpy causes the local maxima to decrease as anticipated. Remarkably, other works have interpreted the non-volatility of OECTs in a similar, though only conceptual way, without direct evidence of the underlying potential function[3,23]. Here, we can demonstrate and prove for the first time the bistability of an OECT that gives rise to its long-term stable, non-volatile behavior. Figure 3f shows the interaction parameters extracted from the data sets of Fig. 3d and e. We find that in all samples, both intraspecies interaction energies ($h_{dd}$ and $h_{uu}$) are of similar small magnitude, and for the samples of high enthalpy (green and blue bars), are strongly outweighed by the interspecies parameter $h_{du}$. For the deliberately

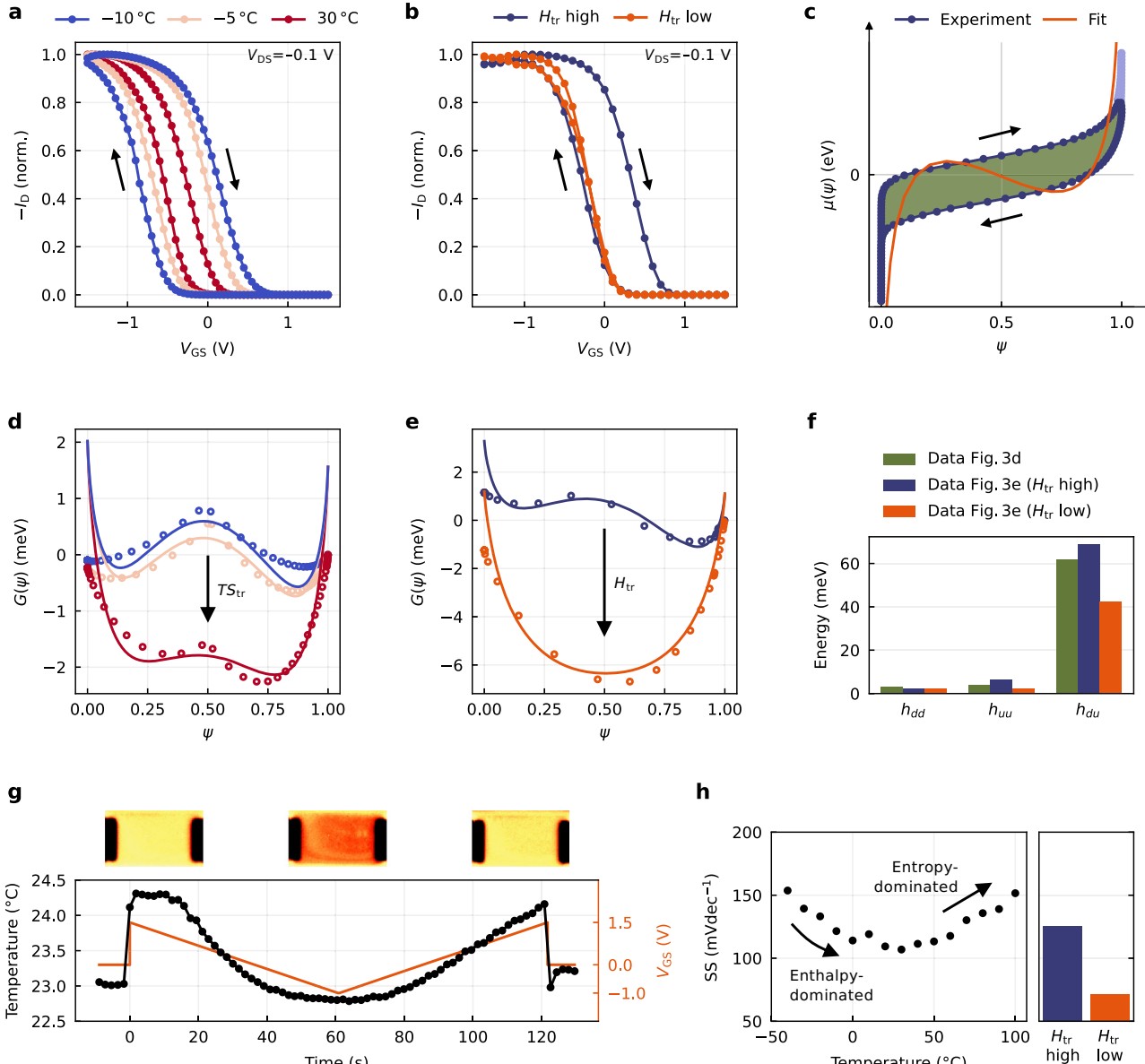

**Fig. 3 | Experimental validation. a, b** Bistability decreases with **a** rising entropy and **b** falling enthalpy ($H_{tr}$ high: untreated device, $H_{tr}$ low: device with added KCl), in line with the expectations from Fig. 2a ($V_{DS} = -0.1$ V). **c** Chemical potential profiles are normalized with symmetry around ($\psi, \mu(\psi)$) = (0.5, 0 eV) and partial integrals of equal size. The analytic continuation was performed to account for the asymmetry of the on- and off-state (light blue data points). The fit corresponds to the chemical potential derived from the fitted Gibbs free energy.

**d, e** Reconstructing and fitting $G(\psi)$ reveals decreasing local maxima for both experiments of **a** and **b**, with the extracted interaction parameters summarized in **f**. **g** Thermal imaging reveals the in- and outflow of heat during device operation, affirming the conclusion from Fig. 2b. **h** The non-monotonic progression of the subthreshold swing with temperature confirms the hypothesis of Fig. 2c. Further, the subthreshold swing decreases for reduced enthalpy as suspected.

modified sample (orange bar), there is a reduction across all parameters, with the most notable decrease in $h_{du}$, underscoring the effective suppression of the interactions between doped and undoped sites. Given these parameters, we can further confirm that both bistable systems fulfill the condition of Eq. (8) with $\lambda > 1$, which decreases with rising temperature and does not apply to the sample of lowered enthalpy, where we find $\lambda < 1$ (Table S2). These findings reinforce the understanding that bistability arises from the balance between entropic and enthalpic interactions involved in the doping cycle. We want to stress though, that the interaction parameters as shown here yet contain another quantity $Z$, which is the coordination number originating from the derivation of the model and which relates to the microstructure of the assumed doping units (Supplementary Note 1). While $Z$ itself is unknown, it is expected a constant in the range typical

for three-dimensional structures (i.e., 2 to ~8) and, therefore, will not change the relative weight of the parameters.

Regarding their physical significance, the $h$-parameters measure the interaction strength of the doping units among and with each other during the doping cycle. In that context, we posit $h_{du}$ to be coupled to the macroscopic transport properties of the semiconductor film, in line with previous reports highlighting the critical role of domain blending and interaction[24,25]. It is reasonable to assume that this blending improves with rising temperature, from which a decreased impact of $h_{du}$ would follow, consistent with the framework presented here (Fig. 3d). Similarly, we observe this parameter to decrease with electrostatic screening (Fig. 3e). We interpret this finding to reflect the situation of OECTs measured with aqueous electrolytes, where typically no bistability is observed. Complementing this, Ji

et al. have shown the significant enhancement of hysteresis by adding low-polar polytetrahydrofuran (PTHF) to a poly(3,4-ethylenedioxythiophene) tosylate (PEDOT:Tos) channel[26]. This finding readily aligns with our model and can be interpreted as follows: Given the aqueous NaCl electrolyte used in their study, a high water content can be expected in the channel, causing little hysteresis in the pristine system due to dielectric screening. That is, enthalpy is suppressed and the system is ruled by entropy. By adding PTHF to the channel, a highly hydrophobic component is introduced, which reduces the exposure to water. It follows a lowered dielectric screening and a rising impact of enthalpy, which gives rise to a distinctive bistability. In this sense, their study appears as a direct counterpart to our experiment of Fig. 3b. Worth highlighting, the authors' microscopic identification of high- and low-resistive regimes involved in charge trapping is a resemblance of the phase transition our model proposes. We expect a similar manifestation for the system studied herein, presumably resulting from the specific interaction between PEDOT:PSS and the ionic liquid[27–29]. In fact, we suspect this phenomenon to extend to a range of organic, non-volatile systems documented in literature[23,26,30–32], under the notion that, despite miscellaneous and system-specific interactions, they share in common that their resulting enthalpic forces exceed entropy.

Our second approach regards the Maxwell construction of Fig. 2b, where the non-monotonic chemical potential is expected to cause a heat exchange. We validate this hypothesis through in-operando thermography studies, which are, to the best of our knowledge, the first of their kind performed at an OECT. As shown in Fig. 3g, we find a heat input equivalent to 1.57 K in temperature difference when switching the device on. Switching off again, we note a decreased flux, indicating slower kinetics for dedoping. These findings confirm our hypothesis of an underlying cycle process, where the performance of chemical work is associated with a heat flux changing sign between the two sweeps. At the same time, however, it is obvious that the OECT's transfer curves lack the sharp transition expected from Fig. 2b. This divergence can be attributed to two factors: First, unlike the idealized two-reservoir model, the actual system is not isolated, but in thermal contact with its environment. Second, the voltage across the channel is not constant and contributes to the slope of the transition, as shown in Supplementary Note 6. Lastly, we want to stress that for the time being, it is challenging to derive further quantities from this experiment, as the precise energetics of the doping reaction underlying this particular OECT system are unknown. While Rebetez et al. have examined the thermodynamics of the doping process of PEDOT:PSS, their study refers to an aqueous electrolyte and should thus not be transferred to the present system without further ado[33]. Nonetheless, our experiment clearly confirms the anticipated heat flux, attributable to the non-monotonic chemical potential.

The third hypothesis finally addresses the subthreshold behavior. We propose that a bistable Gibbs free energy function would manifest itself in a non-monotonic progression of the subthreshold swing with temperature, driven by the balance between enthalpy and entropy (Fig. 2c). In fact, we can prove this exceptional phenomenon experimentally as shown in Fig. 3h (left panel), where the tipping point can be identified at around 30 °C. Expectedly, this trend is accompanied by a continuous decrease in hysteresis, while the transconductance shows a similar non-monotonic progression as the subthreshold swing (Fig. S10). We can further confirm the conjecture of a decrease under reduced enthalpy (Fig. 3h, right panel). As in the former case, this change is driven by a reduced bistability, from which a less pronounced counterbalance from the non-monotonic chemical potential towards the electrostatic potential follows.

## Single-OECT Schmitt trigger
In view of these results, we consider the bistability of the given OECT system as sufficiently substantiated. This finding unlocks a range of promising applications, in particular for the purpose of neuromorphic computing, where the exploitation of hysteresis as a memory function has already been demonstrated thoroughly[3,23,26]. Considering its extent (Fig. 1f), the bistability also appears attractive for another application fundamental in digital and auspicious to neuromorphic electronics: the Schmitt trigger[34,35]. Schmitt triggers are hysteretic devices with separate threshold voltages for rising and falling input. Since the upper and lower output states are only obtained by exceeding these thresholds, suppressive zones are created, making such devices effective noise filters and one-bit analog-to-digital converters (ADCs)[36,37]. In neuromorphic computing, such systems can similarly be leveraged to mimic the behavior of biological neurons, which fire only upon exceeding a certain input threshold[38–41]. Conventionally, however, Schmitt triggers are implemented via comparator circuits, often including at least two transistors and six resistors.

We explore the applicability of the bistable OECT as a single-device Schmitt trigger, using gate and drain as in- and output (Fig. 4a). Shifting a white noise signal in offset for input, the device shows the expected transfer response, while also clearly reflecting the input fluctuations by corresponding output swings (Fig. 4b). Crucially, however, we can identify four distinct regimes of how the noise translates to the output. During the transitions between the on- and off-state, the noise merges noticeably into the output signal (II and IV), whereas in the sections before the transition, there is significant suppression of the same (I and III). In the context of our framework, these transition points appear as being defined by the inflection points $\psi_i$ and $1-\psi_i$, where $\mu(\psi)$ tips into its inverse slope. There, the physical system enters its unstable regime, causing the transfer curve to no longer be determined by the chemical potential alone, and consequently making the output signal more susceptible to a fluctuating input. These regions can be seen even more clearly in Fig. 4c, where the noise has been extracted against an analytical reference (Fig. S11). The even stronger suppression in III compared to I can thereby be attributed to the off-state being susceptible to only one direction of input deflection.

With these results already demonstrating the applicability as a noise filter, the time resolution of the output further demonstrates the use case as an ADC. Taking a noisy triangular function as the input (Fig. 4d, upper panel), the non-steady response allows conversion to bits within a short input window upon exceeding the threshold (Fig. 4d, lower panel). The output states 0 and 1 are each retained over an extended input window despite severe noise. The temporal asymmetry of these states is thereby due to the asymmetric input, to which we bound ourselves for not exhausting the electrochemical window given.

Finally, we analyze the output in the frequency domain by performing the Fourier transformation of an extended switching cycle. Given an underlying bistable potential function, a periodic perturbation is expected to induce nonlinear oscillations that manifest as higher harmonics (Supplementary Note 7). We can confirm that this is the case for the OECT system as well. The fundamental and odd harmonics are clearly visible, as well as several even harmonics with lower amplitude, caused by the non-ideal rectangular shape of the lower half-wave of the output signal. Hence, the signal reveals multiple underlying frequencies, indicating oscillations as expected for a bistable system. To single out one distinct oscillation frequency, an analog or digital filter can be utilized, e.g., a first-order Butterworth filter, of which the spectrum is shown in Fig. 4e.

Taking these results together, it is evident that the OECT's intrinsic bistability allows it to operate as a Schmitt trigger, merging the functionality of a multi-component circuit into a single device. As in other cases[42,43], this is permitted by the key difference between organic electrochemical and conventional devices, namely the use of different charge carrier types and phase systems, inherently allowing for complex operation mechanisms on multiple time scales.

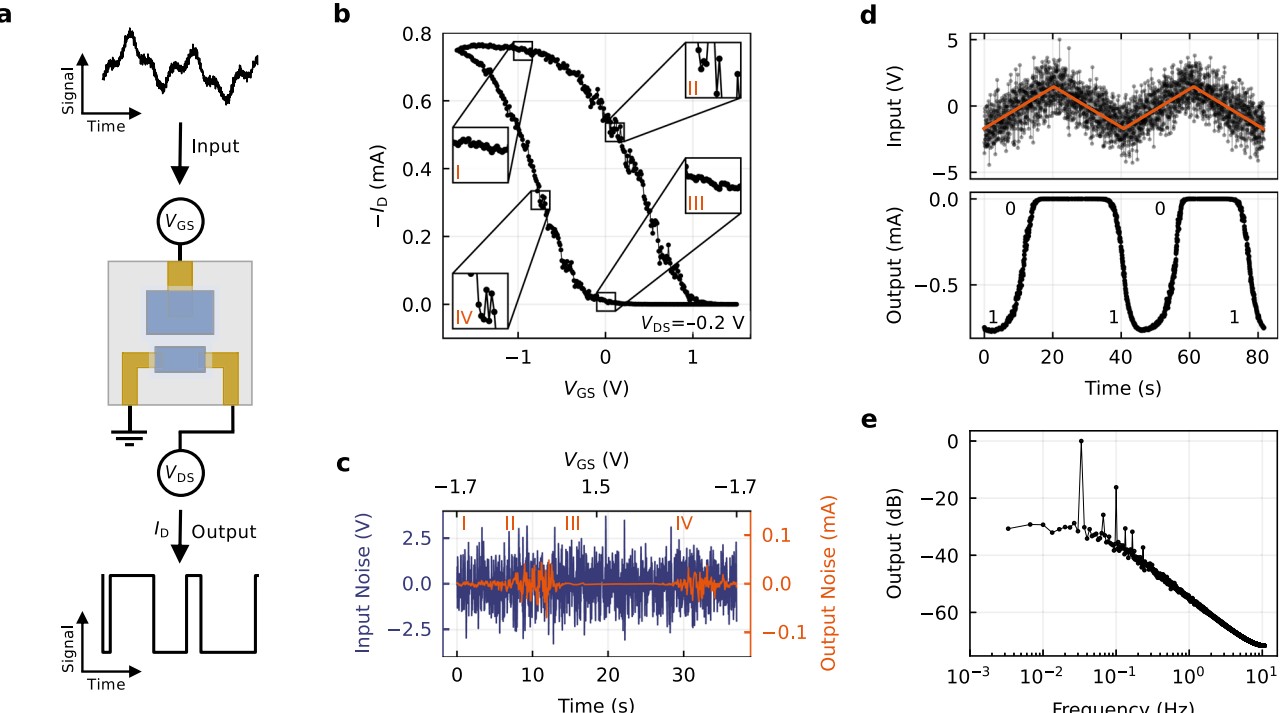

**Fig. 4 | Single-OECT Schmitt trigger. a** The bistable OECT functions as a Schmitt trigger, allowing for noise suppression and analog-to-digital conversion. **b** Transfer curve of an OECT ($V_{DS} = -0.2\,V$) with offset-shifted white noise input signal (Gaussian noise with $f = 400\,Hz$, $V_{pp} = 10\,V$). Insets: Four distinct regimes of output noise. **c** Input vs. output noise as extracted from **b** against an analytical reference (Fig. S11). **d** The device functions as an ADC despite severe input noise. **e** Fourier transformation of a first-order Butterworth filter revealing higher harmonics in the output.

## Discussion

In conclusion, we demonstrate, model, and harness bistable OECTs. We set out with a thermodynamic framework, from which we derive the occurrence of bistability as the consequence of enthalpic effects dominating over entropy. We deduce the consequences this situation would entail and prove these experimentally at an appropriately chosen material system. Among others, we show the exceptional scenario in which the subthreshold swing deviates from Boltzmann statistics. Building on these findings, we demonstrate the functionality of a multi-component circuit with a single OECT device, providing a vital building block for advanced organic circuitry. These insights significantly enhance the understanding of OECT physics and set the stage for their application in non-conventional, complex computing, where bistable systems bear enormous upside[44–46].

## Methods

### Device fabrication

Microfabrication of the OECTs was performed on 1 inch × 1 inch glass substrates, onto which Cr (3 nm) and Au (50 nm) were evaporated for metal traces. AZ 1518 photoresist (MicroChemicals GmbH) was spin-coated at 3000 rpm for 60 s (SAWATEC AG spin-coating system), followed by baking the substrate at 110 °C for 60 s. Metal traces were shaped by illuminating in a maskaligner photolithography system (I-line 365 nm, lamp power 167 W, SÜSS Microtec AG) for 10 s, followed by developing the photoresist in AZ 726 MIF developer (Micro-Chemicals GmbH), and etching Au and Cr for 60 and 20 s (10% diluted aqueous solutions of Standard Gold/Chromium etchants, Merck KGaA). After $O_2$-plasma cleaning, PEDOT:PSS (Clevios PH1000, Heraeus Deutschland GmbH & Co. KG) with 5% v/v ethylene glycol (Merck KGaA) was spin-coated (3000 rpm for 60 s) to yield a PEDOT:PSS layer of ~100 nm. This step was followed by drying at 120 °C for 20 min. Orthogonal photoresist OSCoR 5001 (Orthogonal Inc.) was spin-coated (3000 rpm for 60 s), baked for 60 s at 100 °C, and exposed for 12 s to structure channel and gate. After post-baking (60 s at 100 °C), development followed by covering the sample with Orthogonal Developer 103a (Orthogonal Inc.) and removing it by spinning after 60 s, which was carried out twice. Excess PEDOT:PSS was removed by $O_2$-plasma etching for 5 min with 0.3 mbar (Diener electronic GmbH & Co. KG). Afterwards, the sample was placed in Orthogonal Stripper 900 (Orthogonal Inc.) overnight at room temperature, which was followed by ultrasonic cleaning in ethanol for 15 min. Devices in this work had channel dimensions of $L = 30\,\mu m$, $W = 150\,\mu m$ and featured a PEDOT:PSS side-gate at a distance of 60 μm. For practical reasons, the thermography and Ag/AgCl-gating experiments were carried out with devices of $L = 300\,\mu m$ and $W = 150\,\mu m$. Micrographs are provided in Fig S2.

### Electrolyte preparation and deposition

Preparation and deposition of the solid-state electrolyte was carried out as put forth in ref. 15. The precursor solution was prepared by mixing deionized water (1.0 mL), N-isopropylacrylamide (750.0 mg, Alfa Aesar), N,N'-methylenebisacrylamide (20.0 mg, Merck KGaA), 2-hydroxy-4'-(2-hydroxyethoxy)-2-methylpropiophenone (200.0 mg, Merck KGaA), and the ionic liquid 1-ethyl-3-methylimidazolium ethyl sulfate (1.5 mL, Merck KGaA), followed by stirring overnight at room temperature. Before applying the electrolyte, samples were immersed in a 5% v/v solution of 3-(trimethoxysilyl) propyl methacrylate in buffered ethanol (10% v/v acetic acid/acetate) at 50 °C for 10 min, in order to deposit an adhesion promoting layer. The sample was thoroughly cleaned with ethanol afterwards and dried at 100 °C for 15 min. Deposition of the electrolyte then took place in the maskaligner photolithography system. A drop of precursor solution was put on the devices and carefully covered with a Teflon™ foil, upon which solidification followed by exposing the areas of interest for 20 s through a photomask. Non-crosslinked material was subsequently removed by blowing with an $N_2$-gun. The electrolyte of

the enthalpy-modified systems was deposited by inkjet printing, where the precursor solution was supplemented with 1 mL of ethylene glycol and 2 drops of Triton X-100 to set the viscosity. For modifying the enthalpic contribution, 1 mL of aqueous KCl solution was added. Before application, filtration was carried out through a 0.45 μm PVDF-filter. Placement on the substrate was carried out with a Dimatix Materials DMP-2800 Inkjet Printer, each droplet having a volume of 2.4 pL and a spacing of 15 μm. Solidification took place under UV light for 120 s.

## Electrical characterization

Electrical characterizations were performed in an $N_2$-filled glovebox, with devices sufficiently purged to remove traces of water. Data was acquired using two Keithley 236 SMUs. Temperature-dependent measurements were performed in an $N_2$-filled Janis ST-500 Probe Station with continuous flow cryostat ($LN_2$ cooling), connected to a Scientific Instruments Model 9700 temperature controller and an Agilent HP 4145B. For Schmitt-trigger experiments, a white noise signal was applied through an Agilent 33600A waveform generator. Any setup was controlled by the software SweepMe! (sweep-me.net). All transfer curves were recorded at sufficiently low scan rates ($-10^{-2}\,V\,s^{-1}$) to minimize kinetic effects.

## Thermography measurements

Thermography measurements were performed using an InfraTec ImageIR 9420 thermal camera, installed on a semiautomatic wafer probe station Formfactor CM300. Thus it was possible to align, contact, and measure the samples electrically using single probes while recording the infrared image simultaneously. Electrical biasing and measurements were done via a Keysight B1500 semiconductor parameter analyzer.

## Fits and simulations

Fits and simulations were carried out with customized Python scripts. Gibbs free energy functions were reconstructed by the methods laid out in Supplementary Note 5.

## Data availability

The data underlying the figures in the main text are publicly available from the OPARA repository at https://doi.org/10.25532/OPARA-552. The datasets generated and/or analyzed during the study are available from the corresponding author upon request.

## Code availability

The code underlying the free energy fits is publicly available at https://github.com/lukasbongartz/thermodynamics-fitting.git.

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

## Acknowledgements
L.M.B. and H.K. are grateful for funding from the German Research Foundation (DFG) under the grant KL 2961/5-1 and the Bundesministerium für Bildung und Forschung (BMBF) for funding from the project BAYOEN (01IS21089). H.K. and C.M. acknowledge the project ArNeBOT funded by the Deutsche Forschungsgemeinschaft (DFG, German Research Foundation)—536022519. R.K. thanks the Hector Fellow Academy for support. The authors acknowledge funding from the ct.qmat Cluster of Excellence.

## Author contributions
L.M.B., M.C., and H.K. developed the concept and derived the math. L.M.B. performed simulations. L.M.B., M.C., H.K., and K.L. designed the experiments. A.W. developed the electrolyte. L.M.B. and T.M. carried out device fabrication and transistor measurements. R.H. performed the thermography studies. R.K. and L.M.B. performed the Gibbs free energy fitting and C.M. computed the Butterworth filter. L.M.B. wrote the manuscript. All authors contributed to conceptual discussions and manuscript editing.

## Funding

## Competing interests
The authors declare no competing interests.
