## [Peer Review File · Nature Communications]

Bistable Organic Electrochemical Transistors: Enthalpy vs. EntropyREVIEWER COMMENTS

Reviewer #1 (Remarks to the Author):

Bongartz et al presented recent results of an OCET that shows hysteretic switching behavior between a positive and negative sweep. This hysteresis provided a pathway towards nonvolatile information retention. The authors proposed a thermodynamic model based on two-phase coexistence to explain this nonvolatility, and conducted thermal analysis to verify the two-phase coexistence. This paper is nicely written and well thought out, and the results are significant and truly novel for the field. However, I have three substantial concerns:

First, The experimental methods are incomplete, especially with regards to device fabrication. The authors point to their previous paper (ref. 15), but the methods for each paper should stand on its own. The Methods in the previous paper also did not include details such as the size of the devices, the spacing of the gate and channel electrodes, and other considerations. Moreover, it is not always clear to the audience if the authors slightly changed the recipe or geometry from one paper to another; for this reason, the experimental methods should be re-stated.

The incomplete methods of this work preclude this reviewer from being able to evaluate the primary claims of the work. In particular, the results could make sense if both the gate and channel used the same phase-separating material (PEDOT:PSS). However, it would not be consistent if the gate were made using a metal that act as an electrolyte-gated double-layer capacitor because the voltage of the gate would change even if the voltage of the channel is constant.

The cartoon on Fig. 1d would suggest that the gate is made using an inert metal, while the cartoon on Fig. 4a suggests that the gate is made using a PEDOT:PSS polymer (consistent with ref.15). Fig. S1 contains some optical images, but it is not labeled with the material.

It is recommended that the authors state the entire process flow for device fabrication, and include optical images of the final device where each material is properly labeled. Only then is it possible to evaluate the claims of this work.

Second, the reviewer expresses skepticism that the voltage of OCETs should be described by solution thermodynamics and the chemical potential. Historically, most OCET models have assumed a capacitance-like behavior, where the voltage is given as the charge divided by the capacitance ($Q = C * V$). In their previous work by Cuchhi et al, the authors show that the capacitance model based on electrostatics is insufficient, and that the potential must also incorporate the chemical potential and in particular entropy (equation 9 in Cuchhi). However, the electrostatic component is still clearly present in that equation, and the authors have shown experimental evidence in Fig. 2b,c there that the response of the OCET requires both the electrostatic component [$e V_{gs} - V_{ch}$] and the entropic component.

However, in this work, the authors make a much stronger claim that proposes the enthalpic and entropic components would be stronger than the electrostatic component (line 159). This claim is core to their arguments because both the entropic and electrostatic components would favor single-phase volatile behavior, while the enthalpic components would favor the bistable nonvolatile behavior due to the positive enthalpy of mixing.

From an order of magnitude perspective, this is doubtful. For example, entropic and enthalpic components to the chemical potential would only be about on the order of a few times the thermal voltage (~ 100 mV based on Fig. 1b & 2a of this paper, which is reasonable), and is consistent with other fields like lithium insertion in batteries (Dreyer et al. Nat Mater, 9, 448, 2010; <https://doi.org/10.1038/nmat2730>) and oxygen insertion into strontium cobalt oxide (Lu & Yildiz, Nano letter, 16, 2, 1186; <https://pubs.acs.org/doi/10.1021/acs.nanolett.5b04492>). In these fields, the switching hysteresis is only about tens of millivolts, consistent with what is expected based on mixing enthalpy and configurational entropy.

However, the OCET switching curves show a switching hysteresis of about ± 1 V (Fig. 1f). This would imply an immense positive enthalpy of mixing to overcome the 1V from the electrostatics. The reviewer is unaware of any system with such a large enthalpy of mixing. Indeed, the experimental hysteresis (Fig. 1f, 3a, 3b) are all much larger than the ones given by the

thermodynamic calculations (Fig. 1b, 2a). Here, I'm assuming the Nernst equation with a charge transfer of 1, such that 1 V on the gate = 1 eV chemical potential.

The authors might want to review how phase separation is considered in inorganic versions of electrochemical transistors, also known as electrochemical random-access versions. Examples are (Li et al, ACS Appl. Mater. Int. 11, 38982, 2019; <https://doi.org/10.1021/acsami.9b14338>) and (Kim et al. Adv Electron Mater, Adv. Electron Mater, 2022, 2200958, 2023; <https://doi.org/10.1002/aelm.202200958>)

In addition, there exists a clear picture of electrochemical phase separation in inorganic systems (e.g., the coexistence of two materials with different compositions upon the insertion of ions like lithium and oxygen). However, it is not clear what is "phase separation" in this system other than the thermodynamic picture being drawn with enthalpy and entropy. This reviewer acknowledges that polymers are much harder to characterize, but it also makes the existence of phase separation more speculative.

Third, given the strength of the experimental results in Fig. 1g, it would be better if they can show it can be replicated under different conditions such as temperature in analogy with Fig. 3a.

In summary, I am highly impressed by the nonvolatile information retention shown by the OCET devices in Fig. 1g, especially if it can also be replicated under different conditions at temperatures. However, I am concerned about the thermodynamic interpretation, especially given the inconsistency between the <100 meV predicted hysteresis and the observed hysteresis as high as 1V. That said, I also recognize that new ideas I recognize that new ideas often face initial skepticism, even when they can fundamentally advance the field.

I believe the experimental results (assuming the Methods section is more properly described) will be a valuable addition to the field and deserving of publication. However, the authors should better describe the caveats of the thermodynamic model, and propose other possible explanations for the results.

Reviewer #2 (Remarks to the Author):

The manuscript entitled "Bistable Organic Electrochemical Transistors: Enthalpy vs. Entropy" (Manuscript, No. NCOMMS-24-28128-T) by L. M. Bongartz¹ et al. reports the thermodynamic analysis on bistable OECT devices for a deeper understanding of device physics and application to single-OECT Schmitt triggers. The article covers both theoretical and experimental studies of OECT devices. The analysis in this manuscript was comprehensive and meticulous and has great significance to the field. On the other hand, some parts are missing, such as the justification of the framework of the proposed thermodynamic model. Therefore, I would like to suggest a major revision before publishing in Nature Communications at this moment.

- 1) Do you have any further justification for whether this measurement is being made at the thermodynamic control limit or the kinetic control limit? The scan speed of 10 mV/s is quite slow, but we need to set the scan speed much smaller than the ion injection/extraction time constants. Gate current transients would provide some insight into the validity of the measurement conditions.
- 2) Does this system have no critical temperature in the temperature range of the measurement (especially in Fig3h)? Not only a drastic phase transition like LCST in PNIPAM-water system, but also glass transition or relaxation of side and/or main chains (α, β relaxation) affect thermodynamic functions of the electrolyte and/or channel layer. DSC measurements of solid electrolytes and the PEDOT:PSS-IL system would provide valuable information.
- 3) In the theoretical framework, it is a little difficult to understand the physical and chemical picture of the interaction energy between doped and undoped moieties. Does the interaction energy include the factor of ion to PEDOT electrostatic energy or conformational change upon ion injection?
- 4) It is difficult to follow how to estimate ψ from the transistor data in Figure 3. Is it from the integral of the gate current, assuming the doping efficiency is unity?

Response to the Reviewers

Reviewer #1 (Remarks to the Author):

Bongartz et al presented recent results of an OCET that shows hysteretic switching behavior between a positive and negative sweep. This hysteresis provided a pathway towards nonvolatile information retention. The authors proposed a thermodynamic model based on two-phase coexistence to explain this nonvolatility, and conducted thermal analysis to verify the two-phase coexistence. This paper is nicely written and well thought out, and the results are significant and truly novel for the field. However, I have three substantial concerns:

We thank the reviewer for taking the time to read and examine the manuscript. We very much appreciate your comments and believe the work gained substantial value from your input. Please find below our responses as well as the modifications highlighted in the main text and the supplementary information.

First, The experimental methods are incomplete, especially with regards to device fabrication. The authors point to their previous paper (ref. 15), but the methods for each paper should stand on its own. The Methods in the previous paper also did not include details such as the size of the devices, the spacing of the gate and channel electrodes, and other considerations. Moreover, it is not always clear to the audience if the authors slightly changed the recipe or geometry from one paper to another; for this reason, the experimental methods should be re-stated.

Thank you for pointing that out, we very much agree. We overhauled the method section thoroughly and provide micrographs in the supplementary information for all device types used in this work.

The incomplete methods of this work preclude this reviewer from being able to evaluate the primary claims of the work. In particular, the results could make sense if both the gate and channel used the same phase-separating material (PEDOT:PSS). However, it would not be consistent if the gate were made using a metal that act as an electrolyte-gated double-layer capacitor because the voltage of the gate would change even if the voltage of the channel is constant.

Excellent point. The devices focused herein indeed feature PEDOT:PSS on both channel and gate, except for the experiment where we explicitly operated the device using an Ag/AgCl gate, to confirm the bistability even in that case.

The cartoon on Fig. 1d would suggest that the gate is made using an inert metal, while the cartoon on Fig. 4a suggests that the gate is made using a PEDOT:PSS polymer (consistent with ref.15). Fig. S1 contains some optical images, but it is not labeled with the material.

Thank you for raising that point. We have chosen this generalized OECT scheme so as to direct the focus to the underlying physics instead of the device architecture. As we show, the phenomena described also occur using an Ag/AgCl gate. Nevertheless, we improved in emphasizing that the majority of the experiments were performed with devices of a side-gate architecture.

It is recommended that the authors state the entire process flow for device fabrication, and include optical images of the final device where each material is properly labeled. Only then is it possible to evaluate the claims of this work.

We have extensively revised the methodology and equipped the supplementary information with labeled micrographs.

Second, the reviewer expresses skepticism that the voltage of OCETs should be described by solution thermodynamics and the chemical potential. Historically, most OCET models have assumed a capacitance-like behavior, where the voltage is given as the charge divided by the capacitance ($Q = C * V$). In their previous work by Cuchhi et al, the authors show that the capacitance model based on electrostatics is insufficient, and that the potential must also incorporate the chemical potential and in particular entropy (equation 9 in Cuchhi). However, the electrostatic component is still clearly present in that equation, and the authors have shown experimental evidence in Fig. 2b,c there that the response of the OCET requires both the electrostatic component [$e V_{gs} - V_{ch}$] and the entropic component.

Thank you for that comment. Traditional approaches, such as the model put forth by Bernardis and Malliaras, essentially stem from FET theories and were adopted to OECTs by introducing the volumetric capacitance C^* . It is true that the electrostatic interactions in OECTs are thus typically described with an ideal capacitance. This assumption works in some voltage ranges and provides a decent tool for analysis. However, we showed extensively, and others too (e.g., Fabiano *et al.*, <https://doi.org/10.1039/D3TC03058J>, Keene *et al.*, <https://doi.org/10.1038/s41563-023-01601-5>), that the charge accumulation depends nonlinearly on the voltage applied, for which an ideal capacitor model is insufficient. Similarly, following an ideal capacitor with $Q = C * V$, one would expect the charge Q to increase continuously with the applied voltage V , since the energy to accumulate charges is constant and therefore voltage-independent. For OECTs, this would imply a linear, unrestrained increase in drain current. However, this falls short, as (aside from specific voltage ranges) essentially all OECT systems exhibit a sigmoidal transfer curve with a saturating on-state. This phenomenon can be explained by considering a more fundamental property—the chemical potential—that relates the accumulated charge (*i.e.*, the doping level) to the voltage applied. This gives rise to a “bell”-shaped voltage-dependence of the capacitance, which is a direct insight from our previous work (Cucchi *et al.*, <https://doi.org/10.1038/s41467-022-32182-7>), onto which the present work builds upon. Nonetheless, we would like to further elaborate it in the following for clarification.

First, one can consider the transconductance g_m of an OECT as a function of gate voltage, where $g_m = \partial I_D / \partial V_{GS}$. Given that the drain current I_D depends on the number of holes induced, it can be considered $\propto Q$, for which $g_m \propto C$. This bell-shape dependence of g_m on V_{GS} can be seen in the following figure for the system studied herein (recorded with $V_{DS} = -0.1V$), but in fact, it is ubiquitous throughout OECT literature (e.g., <https://doi.org/10.1038/ncomms3133>, <https://doi.org/10.1002/adma.201303080>).

Second, we measured the bias-dependency of the capacitance directly by means of electrochemical impedance spectroscopy. To this end, impedance spectra were recorded in a range from 0.1 Hz to 10 kHz (small signal amplitude of 10mV) while simultaneously applying a DC-bias in the range from -1.3 to $1.3V$. The capacitance was obtained by a Randles equivalent circuit as commonly used for OECTs (e.g., <https://doi.org/10.1021/acsaelm.3c01673>). As shown in the figure below, this experiment confirms the non-monotonic dependence of the capacitance on the voltage.

These findings are directly in line with our framework. As the number of charges is proportional to the doping parameter ψ , while the electrochemical potential relates to the applied voltage, $C \propto \partial\psi/\partial\mu$, which is just the inverse slope of the chemical potential profile:

However, in this work, the authors make a much stronger claim that proposes the enthalpic and entropic components would be stronger than the electrostatic component (line 159). This claim is core to their arguments because both the entropic and electrostatic components would favor single-phase volatile behavior, while the enthalpic components would favor the bistable nonvolatile behavior due to the positive enthalpy of mixing.

Thank you for explaining your objection. However, we do not consider the electrostatic component to be in competition with the thermodynamic components. Rather, the electrostatic interactions are a consequence of the thermodynamic interactions. We would further like to point out that taking enthalpic contributions into account follows naturally from any thermodynamic consideration, since no system except a hypothetical ideal gas follows exclusively entropic forces. Rather, entropy is always balanced by (internal) energy (*e.g.*, potential energy arising from interactions), which together give rise to the free energy. Given that electrostatic interactions are governed by Coulomb's law, they contribute to the potential (internal) energy of a system and thus impact enthalpy and free energy. It is true that in a number of cases—particularly for systems operating at room temperature and with water-based, strongly screening solvents—entropy is the driving force. Nonetheless, such interactions *can* outweigh entropy under certain circumstances.

For illustration, one may consider a parallel plate capacitor in vacuum or air. The energy required to charge the capacitor is directly related to the electrostatic potential energy. During discharging, this energy is released and can be used to perform useful work or be dissipated as heat, the ratio of which defines the free energy. If the capacitor is now filled with a screening medium, like a salt solution, the plates' charges are neutralized by the ions, reducing the effective electrostatic force between them. Hence, energy is not stored as efficiently, and thermodynamically, this system is ruled by entropy.

We demonstrate the interplay between these two driving forces in the OECT system by means of the subthreshold swing (Fig. 2c, 3h and Supplementary Note 4). This quantity defines the minimum gate voltage required to change

the drain current by one order of magnitude, and in the ideal case, is a purely diffusion-controlled quantity, scaling linearly with temperature. Taking enthalpic interactions into account, however, this process is counterbalanced, and diffusion is impeded. This gives rise to a competition between enthalpic and entropic driving forces, which manifests itself in a non-monotonic temperature-dependence that cannot be accounted for by the entropy-derived Boltzmann's law. It is only for sufficiently risen temperature, when entropy is dominant, that this regime is prevalent.

From an order of magnitude perspective, this is doubtful. For example, entropic and enthalpic components to the chemical potential would only be about on the order of a few times the thermal voltage (~ 100 mV based on Fig. 1b & 2a of this paper, which is reasonable), and is consistent with other fields like lithium insertion in batteries (Dreyer et al. *Nat Mater*, 9, 448, 2010; <https://doi.org/10.1038/nmat2730>) and oxygen insertion into strontium cobalt oxide (Lu & Yildiz, *Nano letter*, 16, 2, 1186; <https://pubs.acs.org/doi/10.1021/acs.nanolett.5b04492>). In these fields, the switching hysteresis is only about tens of millivolts, consistent with what is expected based on mixing enthalpy and configurational entropy.

This is indeed a very good argument. Thank you for providing the chance to settle this point. First, we very much agree with the analogy to the works from the battery community. The resemblance of the (theoretical) energy scales is certainly no coincidence, as it directly follows from the thermodynamics of an ideal gas. However, OECTs involve much more complex coupling mechanisms than inorganic battery systems, which impede this resemblance as we expand on in the next paragraph.

However, the OCET switching curves show a switching hysteresis of about ± 1 V (Fig. 1f). This would imply an immense positive enthalpy of mixing to overcome the 1V from the electrostatics. The reviewer is unaware of any system with such a large enthalpy of mixing. Indeed, the experimental hysteresis (Fig. 1f, 3a, 3b) are all much larger than the ones given by the thermodynamic calculations (Fig. 1b, 2a). Here, I'm assuming the Nernst equation with a charge transfer of 1, such that 1 V on the gate = 1 eV chemical potential.

We agree to this reasoning in essence, but as you pointed out correctly, it assumes a perfect charge transfer efficiency, to which this mismatch is attributed. To elucidate this, let's leave any bistability aside for the moment. Even in that case, when no enthalpy is considered (ideal gas), the thermodynamics take place on the order of a few times the thermal voltage, as you pointed out earlier. However, we see that OECT systems operate at larger energy scales (\sim a few hundred mV to gate between on and off). Hence, the doping efficiency is not perfect, and the Nernst equation is subject to a scaling factor, which physically accounts for the transduction of ionic into electronic currents by OECTs. This mechanism sets OECTs distinctly apart from batteries. By that token, one can also understand how the enthalpic contributions translate to a hysteresis of about ± 1 V in the transfer curve. In fact, we have already addressed this question in Cucchi *et al.* (<https://doi.org/10.1038/s41467-022-32182-7>) by means of a gate factor, but derive it thoroughly in the supplementary information of this work. We apologize if this section has not been emphasized enough. We now devote a separate section to this question, in which we also substantiate it with experimental data (Supplementary Note 6).

The authors might want to review how phase separation is considered in inorganic versions of electrochemical transistors, also known as electrochemical random-access versions. Examples are (Li et al, *ACS Appl. Mater. Int.*

11, 38982, 2019; <https://doi.org/10.1021/acsami.9b14338>) and (Kim et al. Adv Electron Mater, Adv. Electron Mater, 2022, 2200958, 2023; <https://doi.org/10.1002/aelm.202200958>)

Thank you for providing these references. The comparison to inorganic ECRAMs is indeed of value. Especially in view of their track record it offers a perspective for future implementation and scaling of these organic non-volatile systems. With respect to the study of Li *et al.* (<https://doi.org/10.1021/acsami.9b14338>), for instance, we note an interesting parallel in the identification of a minimum subthreshold slope at the boundary of the electrochemically driven phase transformation. This stands in direct analogy to our results in Fig. 2c and 3h, which we believe is a crucial piece of evidence. However, it must be further taken into account that as compared to such inorganic systems, the operation of OECTs involves much more complex mechanisms and irregular environments, where also solvent effects can have a very decisive impact (*e.g.*, polymer swelling). In fact, the authors themselves point out the distinction between their inorganic systems to their polymer-based and organic pendants and note the importance of thermodynamic effects. For example, the study suggests strong electron-electron correlations in Li₂TiO₃-crystals to play a role in the subthreshold swing below the thermodynamic limit. It is questionable that this can be transferred to the substantially more amorphous and rather “wet” OECT systems straightforwardly. Complementing this, Kim *et al.* (<https://doi.org/10.1002/aelm.202200958>) hypothesize that their phase separation occurs between two amorphous states with different metal-to-oxygen ratios that coexist in equilibrium. This is actually quite akin to our microscopic interpretation of the bistability, which is subject to a future study currently in preparation for publication. Interestingly, we notice in that context that the above mentioned study by Dreyer *et al.* (<https://doi.org/10.1038/nmat2730>) proved the thermodynamic treatment of such inorganic systems as viable, which leads us to believe that a similar treatment of inorganic ECRAMs might indeed be feasible as well. For the understanding of OECTs, the thermodynamic treatment allows for a much more generalized discussion of the underlying physics, without precluding any other angles of discussion. For example, as expanded on in the main text, we provide an alternative view on the results of Ji *et al.* (<https://doi.org/10.1038/s41467-021-22680-5>), who reported on the enhancement of OECT hysteresis by the addition of a low-polar solvent. This alternative perspective does not call the authors’ microscopic explanation into question by any means. Rather, it compresses the microscopically manifold effects in the language of thermodynamics, allowing for a broader and much more generalized understanding. Crucially, the authors’ identification of a phase separation into high- and low-resistive regimes involved in charge trapping is in direct agreement with the phase separation our model proposes, and foreshadows our microscopic interpretation for the present system alike. We have updated the main text in that respect.

In addition, there exists a clear picture of electrochemical phase separation in inorganic systems (*e.g.*, the coexistence of two materials with different compositions upon the insertion of ions like lithium and oxygen). However, it is not clear what is “phase separation” in this system other than the thermodynamic picture being drawn with enthalpy and entropy. This reviewer acknowledges that polymers are much harder to characterize, but it also makes the existence of phase separation more speculative.

We agree that phase separation in inorganic systems is much more tangible than it is for polymers, not least because it is subject to a variety of interdependent factors (*e.g.*, polymer composition, electrolyte, solvents). In fact, this is a core strength of our thermodynamic treatment, since it allows to describe the state of the system in a holistic manner as a function of the doping level ψ . This also forms the rationale for our studies based on

thermodynamic variables (Fig. 2 and 3). Nevertheless, we very much understand the desire for a microscopic look into the material, which is the subject of a study currently being prepared for publication. Further, as noted above, we improved on discussing that matter in the main text.

Third, given the strength of the experimental results in Fig. 1g, it would be better if they can show it can be replicated under different conditions such as temperature in analogy with Fig. 3a.

Thank you for that comment. We show the replication of this experiment in the supplementary information (Fig. S8, S9), which confirms the translation from a reduced hysteresis to state retention. We further include a figure that sets the entropy- and enthalpy-dominated state retention in direct comparison.

In summary, I am highly impressed by the nonvolatile information retention shown by the OCET devices in Fig. 1g, especially if it can also be replicated under different conditions at temperatures. However, I am concerned about the thermodynamic interpretation, especially given the inconsistency between the <100 meV predicted hysteresis and the observed hysteresis as high as 1V. That said, I also recognize that new ideas I recognize that new ideas often face initial skepticism, even when they can fundamentally advance the field.

We very much appreciate this note, and we do see discourse as a core element to scientific advancement. Particularly with respect to the emphasized discrepancy between the predicted and experimentally observed hysteresis, we hope that our explanations and derivations will bring clarity.

I believe the experimental results (assuming the Methods section is more properly described) will be a valuable addition to the field and deserving of publication. However, the authors should better describe the caveats of the thermodynamic model, and propose other possible explanations for the results.

As can be seen in the main text, we have fundamentally revised the experimental methods and expanded the supplementary information accordingly. We have also updated the main text with regard to the microscopic manifestation of our model. We hope that our comments and data will provide the needed clarification. Thank you for your efforts.

Reviewer #2 (Remarks to the Author):

The manuscript entitled "Bistable Organic Electrochemical Transistors: Enthalpy vs. Entropy" (Manuscript, No. NCOMMS-24-28128-T) by L. M. Bongartz1 et al. reports the thermodynamic analysis on bistable OECT devices for a deeper understanding of device physics and application to single-OECT Schmitt triggers. The article covers both theoretical and experimental studies of OECT devices. The analysis in this manuscript was comprehensive and meticulous and has great significance to the field. On the other hand, some parts are missing, such as the justification of the framework of the proposed thermodynamic model. Therefore, I would like to suggest a major revision before publishing in Nature Communications at this moment.

Thank you for taking the time to examine the manuscript. We greatly appreciate your input and have put according revisions into place. Please find our responses below as well as the revisions highlighted in the main text and the supplementary information. Thank you for taking the time.

Do you have any further justification for whether this measurement is being made at the thermodynamic control limit or the kinetic control limit? The scan speed of 10 mV/s is quite slow, but we need to set the scan speed much smaller than the ion injection/extraction time constants. Gate current transients would provide some insight into the validity of the measurement conditions.

Very good point. Indeed, the already little scan speed of 10 mV/s can still be lowered to absolutely rule out any kinetic effects. We have addressed this question previously in two publications prior to this work (Weissbach, Bongartz *et al.*, J. Mater. Chem. C, 2022, 10, 2656-2662, <https://doi.org/10.1039/D1TC04230K> and Shameem *et al.*, Applied Sciences 13, 5754 (2023), <https://doi.org/10.3390/app13095754>), where we show that the hysteresis of this particular OECT system does in fact not vanish even for scan rates orders of magnitude lower. Rather, it converges to a minimum degree, indicative of a thermodynamic bistability. In these publications, we discuss these two "hysteresis regimes" as kinetic and non-kinetic (see for instance the second publication, Fig. 2b). To illustrate this further, please find below the data of a transfer sweep we recorded over a course of 9 h, with a scan rate as little as 180 μ V/s.

In addition, we show state retention transients including the gate current, showing the stable non-volatility.

We have also included these plots in the supplementary information, which we believe is an important addition.

Does this system have no critical temperature in the temperature range of the measurement (especially in Fig3h)? Not only a drastic phase transition like LCST in PNIPAM-water system, but also glass transition or relaxation of side and/or main chains (α, β relaxation) affect thermodynamic functions of the electrolyte and/or channel layer. DSC measurements of solid electrolytes and the PEDOT:PSS-IL system would provide valuable information.

Thank you for that question, that's a good remark. We could not yet identify any signs for such a critical phase transition, which might particularly be attributed to the fact that we operate all devices in an inert atmosphere and only after thorough purging in vacuum. Further, we find the bistability to be independent of the PNIPAM solid-state matrix, and to similarly occur when the ionic liquid is applied as a free-standing electrolyte. But indeed, DSC studies of the PEDOT:PSS-IL system is a valuable suggestion, which we consider for our following microscopic material studies.

In the theoretical framework, it is a little difficult to understand the physical and chemical picture of the interaction energy between doped and undoped moieties. Does the interaction energy include the factor of ion to PEDOT electrostatic energy or conformational change upon ion injection?

Yes, it does: The transfer curve of the OECT is subject to any of these effects, therefore the Gibbs free energy—and thus the interaction parameters—are alike. This is a key motivation behind the thermodynamic treatment, as it encapsulates any interactions taking place into macroscopic state variables. However, as is true for any statistical treatment, this holistic view goes along with the drawback of limited insight into the microscopic origin of the same—at least without further ado, which we follow on in a study currently being prepared for publication. Nevertheless, we have expanded the main text in this respect in order to also address the question of a microscopic interpretation.

It is difficult to follow how to estimate ψ from the transistor data in Figure 3. Is it from the integral of the gate current, assuming the doping efficiency is unity?

The doping parameter ψ is grounded in the notion that the saturated on-current corresponds to the situation where all available doping sites are doped ($\psi = 1$), while the reverse is true for the saturated off-state ($\psi = 0$). In that sense, ψ is straightforwardly estimated by means of the normalized drain current. The integration of the transfer curve then allows to calculate the free energy profile as a function of ψ , from which the doping efficiency is inferred (see Supplementary Note 6 for a detailed derivation).

REVIEWERS' COMMENTS

Reviewer #1 (Remarks to the Author):

The authors have addressed my concerns. I still have doubts about the magnitude of the hysteresis and the proposed origins for the enthalpy. However, this does not diminish the substantial impact of the work and the excellent theoretical and experimental foundations. I believe the publication of this work would enable this field to move forward in the correct direction.

A minor comment is that I hope the authors can better distinguish which plots are experimental and which ones are theoretical derivations, especially when the I_d - V_g curves look similar. Fig. 3 and 4 are clearly labeled as experiments, but Fig. 2 was not labeled as theoretical. Fig. 1 has a mix of experimental and theoretical curves which could make it more challenging for readers if it is not clearly labeled.

Reviewer #2 (Remarks to the Author):

After this revision, the manuscript has been improved. Also, the authors' comments contain very good points to be described in the manuscript to clarify the content of the paper. I would suggest some minor corrections before accepting this manuscript:

1) The description of the theoretical background has been improved. Supplementary Note 6, especially Equation S43, helps to understand the overall theoretical framework. Does it mean the temperature dependency of doping share comes from the same origin as the Poisson–Boltzmann equation on diffusive double layers? If so, the discussion with the other reviewer would be simplified and more straightforward.

2) I understand the authors' motivation for setting the framework of the interaction energy: to provide a macroscopic view of the doping process. The Flory-Hugging theory is an old-school theory, and it fits to describe the phenomena, but the subscripts H_{mix} and S_{mix} would be misleading because the original Flory-Huggins is a theory for mixing two polymers (or polymers and solvents), whereas this manuscript discusses the doping/dedoping reaction without changing the phase-separated structure. Some other subscripts, like H_{doping} , may be better for readers.

Response to the Reviewers

Reviewer #1 (Remarks to the Author):

The authors have addressed my concerns. I still have doubts about the magnitude of the hysteresis and the proposed origins for the enthalpy. However, this does not diminish the substantial impact of the work and the excellent theoretical and experimental foundations. I believe the publication of this work would enable this field to move forward in the correct direction.

We thank the reviewer for taking the time to evaluate the revised manuscript and we very much appreciate the recognition.

A minor comment is that I hope the authors can better distinguish which plots are experimental and which ones are theoretical derivations, especially when the Id-Vg curves look similar. Fig. 3 and 4 are clearly labeled as experiments, but Fig. 2 was not labeled as theoretical. Fig. 1 has a mix of experimental and theoretical curves which could make it more challenging for readers if it is not clearly labeled.

Thank you for that comment. We have added corresponding remarks where they were previously missing (Fig. 1 and 2).

Reviewer #2 (Remarks to the Author):

After this revision, the manuscript has been improved. Also, the authors' comments contain very good points to be described in the manuscript to clarify the content of the paper. I would suggest some minor corrections before accepting this manuscript:

We thank the reviewer for his efforts in evaluating the revised manuscript, and we appreciate the recognition of our adjustments.

1) The description of the theoretical background has been improved. Supplementary Note 6, especially Equation S43, helps to understand the overall theoretical framework. Does it mean the temperature dependency of doping share comes from the same origin as the Poisson–Boltzmann equation on diffusive double layers? If so, the discussion with the other reviewer would be simplified and more straightforward.

This is a very good question, thank you. Indeed, it is reasonable to recognize the temperature dependence stemming from the Poisson-Boltzmann equation. However, we see that this is not sufficient to capture the phenomena focused herein entirely. For example, the doping efficiency appears to be linearly dependent on temperature (Fig. S17), while the hysteresis reveals a non-monotonic dependency on temperature (Fig. 3a). A similar argument follows for the subthreshold behavior (Fig. 3h). Therefore, for the time being, we treat these factors separately and consider the doping efficiency and its temperature dependence rather as a fudge factor.

2) I understand the authors' motivation for setting the framework of the interaction energy: to provide a macroscopic view of the doping process. The Flory-Hugging theory is an old-school theory, and it fits to describe

the phenomena, but the subscripts H_{mix} and S_{mix} would be misleading because the original Flory-Huggins is a theory for mixing two polymers (or polymers and solvents), whereas this manuscript discusses the doping/dedoping reaction without changing the phase-separated structure. Some other subscripts, like H_{doping} , may be better for readers.

Thank you for that remark, this is indeed very valuable. We now changed the subscripts to “tr”, indicating the entropy and enthalpy involved in the state transition during doping and dedoping.